# Study on Surface Integrity and Fatigue Properties of TC4 Titanium Alloy by Surface Ultrasonic Rolling

**DOI:** 10.3390/ma16020485

**Published:** 2023-01-04

**Authors:** Xiaotong Zhu, Pengtao Liu, Chi Zhang, Hao Liang, Jun Hua

**Affiliations:** School of Materials Science and Engineering, Dalian Jiaotong University, Dalian 116028, China

**Keywords:** TC4 titanium alloy, surface ultrasonic rolling, surface integrity, fatigue property

## Abstract

In this paper, the influence of a surface ultrasonic rolling process on the surface integrity of TC4 titanium alloy and its influence on the fatigue properties were studied. By comparing and analyzing the surface roughness, microhardness, residual stress, microstructure, and fatigue fracture, the surface strengthening and modification mechanism of TC4 titanium alloy is discussed. The results show that the surface roughness of titanium alloy is observably decreased after the suitable surface ultrasonic rolling process, and the maximum Ra value can be reduced to 0.052 μm. The axial residual stress on the specimen surface can be increased to −685 MPa. The hardening rate of the surface hardness of the sample was 35%. The residual compressive stress and hardness of the sample surface increased with the increase of static pressure. However, the increase of feed rate and rational speed was less. After surface ultrasonic rolling, the sample surface exhibited obvious grain refinement, the number of high-angle boundaries increased to include the formation of nano-equiaxed grains. The fatigue strength increased by 52% from 280 MPa to 425 MPa. Under 450 MPa, the fatigue life of samples with SUR 2 was the highest, at about 7.7 times that of the original samples. The surface integrity of titanium alloy samples after surface ultrasonic rolling treatment is greatly improved, which is the reason for the significant increase in fatigue life of the samples.

## 1. Introduction

Titanium alloy is an important structural material with high strength, low density, and excellent corrosion resistance, and easily meets light weight requirements. It is widely used in automotive, aerospace, biomedical, and other fields [1,2,3]. However, it has the disadvantages of low hardness, low wear resistance, and low fatigue resistance [4,5,6]. If titanium alloy is too light, parts will be damaged, and if it is too heavy, major safety accidents will occur, which limits its application range.

The surface machining technology of titanium alloy has been studied extensively. Sartori studied the integrity of the semi-finished surface of Ti6Al4V titanium alloy under different cryogenic cooling techniques, and the results showed that the machined surface had no defects and had the best integrity when N_2_ cooling was used at −150 °C [7]. Lin treated the surface of titanium alloy with sandblasting and grinding methods and carried out 3-point bending test. The results showed that the Ti–ceramic bond strength of sandblasting samples was higher than that of grinding samples [8]. Gupta used plasma spraying technology to prepare TiN coating on a Ti6Al4V surface in situ, and the results showed that the coating showed unusually high wear- and corrosion- resistant properties. After spraying, the wear rate of Ti6Al4V was significantly reduced by about 15 times, and the corrosion rate was significantly reduced by about 4 times [9]. Lv conducted an ultrasonic deep rolling (UDR) treatment on the surface of titanium alloy, and studied the influence of UDR on the surface morphology and surface roughness, showing that UDR treatment can significantly reduce machining marks and greatly reduce the surface roughness [10]. Li used shot peening to carry out a surface-strengthening treatment on TC11 titanium alloy, showing that shot peening can introduce a residual compressive stress field with thickness of about 230 MPa on the surface layer of the sample [11]. Xue adopted the method of micro-laser shock strengthening to strengthen the TC17 titanium alloy, and the results showed that the fatigue strength of the sample increased by 32% [12]. Kattoura used ultrasonic nanocrystal surface modification (UNSM) technology to surface process ATI 718 Plus. The results showed that after UNSM, nano-sized grains, twins, and high fault density were generated in the near-surface region of the sample, and the increase in surface hardness and residual stress significantly reduced the crack growth rate [13]. Ren used different ultrasonic surface rolling (USR) processes to roll Ti5Al4Mo6V2Nb1Fe titanium alloy. The results showed that serious plastic deformation occurred on the sample surface, and the surface roughness, hardness, and residual stress were optimized. After USR, the wear rate was reduced by about 70%, and the fretting wear resistance was improved [14]. Kumar used ultrasonic shot peening to treat the surface of titanium alloy, and the results showed that a deformation layer of about 30 μm was formed on the surface of the sample, with a grain size of 17–25 nm and significant refinement [15].

To sum up, most scholars use low-temperature cooling, shot peening, ultrasonic surface rolling and other technologies to treat the surface of titanium alloy, so as to improve surface properties, while the use of surface ultrasonic rolling technology to study the surface integrity and microstructure of titanium alloy changes is not deep enough. In this paper, a surface ultrasonic rolling process was used to study TC4 titanium alloy surface ultrasonic rolling. The influence of ultrasonic rolling process parameters on surface properties was studied by changing static pressure, feed rate, and rational speed, and the influence rule on fatigue properties of titanium alloy materials was studied. The surface modification and fatigue properties of TC4 titanium alloy were studied in depth, with the aim of providing the reference basis and theoretical support for the anti-fatigue manufacturing of TC4 titanium alloy.

## 2. Test Details

### 2.1. Experimental Materials and Methods

The material selected for the test was TC4 titanium alloy with α + β dual-phase structure and a hardness of 315 HV. The main chemical composition is shown in Table 1. The original material was annealed TC4 titanium alloy sheet with 20 mm thickness, and its main chemical composition is shown in Table 1. The TC4 titanium alloy sheet is cut and machined to obtain fatigue samples. The processing size is shown in Figure 1.

The H^+B^ 6063 surface ultrasonic rolling equipment (Huayun, Jinan, China) is used to carry out ultrasonic rolling treatment on the surface of the turning sample, as shown in Figure 2. Test parameters: vibration frequency 29 kHz, current 1 A, amplitude 0.007–0.008 mm, rolling ball diameter of 14 mm. The three variables are rational speed, feed rate, and static pressure, respectively. The variation range of rational speed is 10–1000 r/min, feed rate is 0.1–0.6 mm/r, and static pressure is 0–350 kg. The single factor variable method is used to carry out the surface ultrasonic rolling process test. The test parameters selected for the test are shown in Table 2, and the macro morphology of the treated samples is shown in Figure 3.

The fatigue sample is mounted on the PQ-6 rotary bending fatigue test machine (Qingshan, Yinchuan, China). The rotational speed of the equipment is 3000 r/min, which ends when the sample breaks or the revolution reaches 1 × 10^7^. The rotary bending fatigue test is carried out by the lifting method, and 5 samples are made under each stress. The fatigue life under each stress is arranged from long to short, and the middle two values with the smallest difference are taken as the fatigue life under this process. The S-N curve is obtained by fitting each point.

The surface roughness (Ra) of samples before and after surface ultrasonic rolling is measured with a JD520 surface roughness meter (Jitai Keyi, Beijing, China). The surface texture and the thickness of the deformation layer are observed using a LEICA DCM3D microscope (Leica Microsystems, Heidelberg, Germany). The residual stress on the surface of the samples is measured with an i-XRD X-ray stress tester (Proto Company, Windsor, Canada). The type of tube ball is Co, the test voltage is 25 kV, the operating current is 4.5 mA, and the diffraction angle is 155°. The angle instrument is used for accurate measurement, the exposure time is 1 s, the spot size is 2 mm × 1 mm, the diffraction crystal plane is (114), the scanning range of Bragg angle 2θ is 120°~170°, and the peak location method is Gaussian. The microhardness is measured by a FM-700 hardness tester (Future-Tech Corporation, Kawasaki, Japan), the measuring load is 10 gf, and the loading time is 15 s. A SUPRA55 field emission scanning electron microscope (FEG-SEM, Carl Zeiss, Jena, Germany) is used to observe the fracture morphology changes of the sample after fatigue failure. The accelerated voltage is 15 kV. Working distance (WD) is 9–10 mm. EBSD data collection is performed using a NordlysMax2 detector and Aztec software (3.1 version, Oxford Instruments, Oxford, UK), whereas a Channel 5 software package (5.1120450.0 version, Oxford Instruments, Oxford, UK) is used for the post-processing of the data (all from Oxford Instruments Inc.). The acceleration voltage is 20 kV, the working distance (WD) is 14 mm~16 mm, the sample platform tilt is 70°, and the step size is 400 nm. The surface microstructure of samples is analyzed with a JEM-2100F (JEOL Ltd., Tokyo, Japan) transmission electron microscope (TEM). 

### 2.2. SEM, OM, EBSD, and TEM Sample Preparation

The fractured sample is cut into small samples with a height of 20 mm and a diameter of 9.5 mm along the fracture direction. Petroleum ether and alcohol are sequentially used for ultrasonic cleaning in the SB-5200DTN ultrasonic cleaner (Scientz, Ningbo, China) to prevent oxidation of the fracture. Qualified fracture samples are prepared, and then the fracture morphology is observed by SEM.

All OM and EBSD samples are cut into 8 mm × 8 mm × 3 mm along the rolling direction and then progressively polished with 400–1200 grit SiC sandpaper [16]. Finally, the samples are polished with 10 μm, 5 μm, 3 μm, and 1μm diamond paste for 6 min, 5 min, 5 min, and 5 min, respectively. The corrosion solution is prepared using 5% HF + 10% HNO_3_ + 85% H_2_O (distilled water), and the polished samples are etched to prepare qualified OM samples. The GATAN Ilion II 697 (Gatan, CA, USA) argon ion polishing equipment is used to polish the sample surface with argon ions; the voltage is 5 kV and the polishing time is 4 h, and qualified EBSD samples are prepared.

The TEM sample is prepared by using a cutting machine (Huafang, Hangzhou, China) to cut a 3.2-mm-diameter semi-cylinder from the section of the sample, using a low-speed saw to cut it into a 1-mm-thick semi-circular slice, polishing it into a 35–40-μm semi-circular slice with different types of sandpaper, and then using a Gatan PIPS 691 ion (Gatan, CA, USA) thinning instrument for ion thinning of the slice. TEM samples are prepared.

## 3. Results

### 3.1. Surface Roughness and Surface Texture

After different surface ultrasonic rolling processes, the surface roughness value (Ra) and the reduction rate of roughness were obtained, as shown in Figure 4. The surface roughness of the original sample was about 1.892 μm. After surface ultrasonic rolling treatment, the surface roughness of the samples changed by between 0.05 μm and 0.2 μm, and the reduction rate of the roughness value was 88% to 97.2%. It was found that after the surface ultrasonic rolling treatment, the surface roughness values of the samples decreased significantly, and the reduction rate of roughness increased. In comparing SUR 1, SUR 7, and SUR 2 samples, it was noted that the surface roughness observably decreased with the increase in static pressure. In comparing SUR 3, SUR 7 and SUR 4, SUR 5, and SUR 7 and SUR 6 samples with the increase in feed rate and rotational speed, it was noted that the roughness difference decreased and the roughness reduction rate also decreased.

The three-dimensional morphology of the sample surface under different processes was characterized, as shown in Figure 5. The original sample had obvious convex peaks and concave valleys caused by turning (Figure 5a), and the distribution of convex peaks and concave valleys was relatively uniform, and the height difference between the highest position of the convex peak and the lowest position of the concave valley was 13 µm. After surface ultrasonic rolling treatment, the sample surface became relatively flat and the convex peaks were flattened (Figure 5b–h), among which it was found, in process SUR 7, that the surface treatment effect was the best.

### 3.2. Surface Residual Stress Analysis

The distribution of residual stress on the sample surface after different surface ultrasonic rolling processes is shown in Figure 6. The surface residual stresses of the original samples were axial (TD) −260 MPa and circumferential (RD) −45 MPa, respectively. After surface ultrasonic rolling treatment, the residual stress in the TD direction and the RD direction increased, and the residual stress in the TD direction was higher than that in RD direction. When the static pressure increased, the residual compressive stress in the direction of SUR 1, SUR 7, and SUR 2 increased from −462 MPa to −685 MPa, and the residual stress in the direction of RD increased from −184 MPa to −216 MPa. Therefore, the static pressure increased and the residual stress increased. With the increase of the feed rate, the residual stress in the TD direction of SUR 3, SUR 7, and SUR 4 decreased from −600 MPa to −576 MPa, and the residual stress in the RD direction decreased from −181 MPa to −153 MPa. When the rotational speed of SUR 5, SUR 7, and SUR 6 increased, the residual stress in the TD direction decreased from −600 MPa to −513 MPa, and the residual stress in RD direction decreased from −159 MPa to −143 MPa.

### 3.3. Changes in Microhardness and Microstructure

The surface hardness and hardening rates of the samples after different surface ultrasonic rolling process treatments are shown in Figure 7a. Formula (1) is usually used to calculate the hardening rate. The surface hardness of the original sample (OR) was approximately 315 HV. After surface ultrasonic rolling treatment, the surface hardness of the sample increased to different degrees, and the sample surface was hardened to different degrees. When the static pressure increased from 30 kg to 90 kg, the hardness increased from 350 to 425 HV, and the hardening rate increased from 11% to 35%. When the feed rate increased from 0.05 mm/r to 0.15 mm/r, the hardness increment decreased and the hardening rate decreased from 30% to 8%. Therefore, when the feed rate was 0.05 mm/r, the surface hardness was 410 HV, and the hardening effect was better. When the rational speed increased from 28 to 132 r/min, the hardening rate decreased from 28% to 5%. Therefore, when the rational speed was 28 r/min, the hardening effect was relatively good and the hardness was 390 HV.
(1)$$Hardening\;rate=\frac{{HV}_{after}-{HV}_{original}}{{HV}_{original}}$$
The cross-sectional hardness changes of the samples after different surface ultrasonic rolling process treatments are shown in Figure 7b. After surface ultrasonic rolling treatment, the cross-sectional hardness of the samples was increased, and the surface to the matrix showed a decreasing trend. The samples of SUR 1, SUR 7, and SUR 2 were compared, and the hardness of the cross section observably increased with the increase in static pressure. The samples of SUR 3, SUR 74 and SUR 4, SUR 5, SUR 7, and SUR 6 were compared. With the increase in feed rate and rotational speed, the hardness was relatively reduced. The SUR 1–SUR 7 hardened layer thicknesses were measured as 20 μm, 40 μm, 25 μm, 10 μm, 25 μm, 10 μm and 20 μm, respectively. The same phenomenon occurred when the static pressure changed in the thickness of the hardened layer, mainly because the distance between the two hardness points was large during the hardness test, which crossed the hardened layer. The hardening layer thickness decreased with the change of feed rate and rotational speed.

The surface microstructure changes of samples under different surface ultrasonic rolling processes are shown in Figure 8. Compared with the original microstructure (Figure 8h), plastic deformation of different degrees occurred in the surface layer after surface ultrasonic rolling treatment. With the increase in static pressure, the thickness of the deformation layer increased, and the thickness of the plastic deformation layer in SUR 2 was the largest, at about 40 μm. With the increase in feed rate and rotational speed, the thickness of the plastic deformation layer decreased.

Surface microstructure of SUR 2 samples after surface ultrasonic rolling was characterized by EBSD, as shown in Figure 9. All the EBSD maps were first cleaned by removing wild orientation spikes and filling in zero solutions via extrapolation of up to six neighbours. Among them, the inverse pole figure Y-axis (IPFY) in colour for a visualisation of the structure of lattice domains (grains). Grain boundary distribution maps indicate the number of grain boundaries with different angles. In this study, misorientations between 2° and 10° were considered to be low-angle boundaries, whereas those greater than 10° were high-angle boundaries. It can be observed that the surface layer had obvious grain refinement, while the original grain was coarse. Twins were also found in the large inner grains (Figure 9a–c). The reason is that the α phase in TC4 titanium alloy is a densely packed hexagonal structure. Since there are few slip systems in the densely packed hexagonal structure, it is difficult for the polycrystal to slip when plastic deformation occurs. When the stress is large, the plastic deformation will be in the form of twin crystals. With the increase in deformation, dislocation within the titanium alloy grains gradually increased and the grains became entangled, forming low-angle boundaries (Figure 9d,e). When the deformation continued to increase, the low-angle boundary inside the grain was gradually transformed into a high-angle boundary (Figure 9e), which further refined the surface grain.

The TEM characterization of the microstructure changes of SUR 2 samples at different distances from the surface is shown in Figure 10. At 20 μm from the surface, the deformation was small and a large number of dislocations moved, leading to the dislocations piling up (Figure 10d). At 15 μm from the surface, the deformation increased, the grains were elongated into lamellar shapes, the dislocation density increased significantly, and the grains were refined (Figure 10c). As the distance from the surface decreased, the deformation further increased, the lamellar spacing decreased, and the grain refinement increased (Figure 10b). Equiaxed nanocrystalline structures were formed about 1.5 μm into the surface layer (Figure 10a).

### 3.4. Fatigue Property

#### Fatigue Life

According to the test results for surface roughness, hardness, microstructure, and surface residual stress, four groups of samples treated by surface ultrasonic rolling process were selected to carry out rotary bending fatigue testing, and 6 samples are used in each process. The rotational speed was 3000 r/min, and the contact stress was 450 MPa. Figure 11 shows the fatigue life of the original and surface ultrasonic rolling treatment samples. The fatigue life of samples treated by different surface ultrasonic rolling processes was improved to different degrees. The average fatigue life of the original sample was 0.71 × 10^6^ cycles, and the fatigue lives of SUR 1, SUR 2, SUR 6, and SUR 7 were 3.02 × 10^6^ cycles, 5.48 × 10^6^ cycles, 4.03 × 10^6^ cycles, and 2.06 × 10^6^ cycles, respectively. It was found that the fatigue life of SUR 2 was the highest, at about 7.7 times that of the original sample.

In order to further study the S-N curves of the fatigue life of the original and SUR 2 samples, the original and SUR 2 samples were selected to carry out fatigue testing with four groups of stress levels, and six samples were used for each stress level. Figure 12 shows the S-N curves of the original and SUR 2 samples. The original sample did not have fatigue fracture after 1 × 10^7^ cycles at 280 MPa. With the continuous increase in stress, the original sample experienced fatigue fracture at the stress levels of 300 MPa, 320 MPa, and 340 MPa, and the fatigue life of the sample decreased with the continuous increase in stress level.

No fatigue fracture occurred in SUR 2 samples after 1 × 10^7^ cycles at 425 MPa. Under this process, the fatigue life was relatively high. With the increase in stress, fatigue fracture occurred in the SUR 2 samples under the stress levels of 450 MPa, 475 MPa, and 500 MPa. With the continuous increase in stress levels, the fatigue lives of the samples gradually decreased. It could also be found that the fatigue stress of the sample after surface ultrasonic rolling treatment was higher than that of the original sample.

### 3.5. Fatigue Fracture Analysis

The fracture morphology of all samples in this paper was composed of three regions: the crack source region, the crack propagation region, and the fracture region. Figure 13 shows the fatigue fracture morphology of the original sample at 450 MPa. The original sample cracks started at the surface and then propagated into the material (Figure 13a,b). However, obvious cleavage could be observed in region (c), and the crack was propagated by plastic fracture, indicating that region (c) was the crack propagation region. Therefore, brittle fracture occurred in the stage of the crack source and propagation, and the crack propagation was carried out by plastic fracture. With the increase in operation cycles, cracks propagated rapidly, resulting in instantaneous fracture of the material, and obvious dimple morphology existed on its surface Figure 13d).

The fatigue fracture morphology of the SUR 2 sample at 450 MPa is shown in Figure 14. After surface ultrasonic rolling treatment, the fatigue crack of the sample was generated at the subsurface (Figure 14a,b) and then propagated around, so there was always cleavage in the propagation region (Figure 14c). As most sections of the sample were fractured, the unfractured part could not support the action of external stress and would break instantaneously, forming the morphology of dimples (Figure 14d).

## 4. Discussion

### 4.1. Influence of Surface Ultrasonic Rolling on Surface Integrity

In the process of surface ultrasonic rolling treatment, the static pressure, feed rate and rotational speed directly determine the surface machining degree of the sample. After surface ultrasonic rolling treatment, the metal on the surface of the sample will undergo plastic deformation, which will prompt the material with convex peak to flow rapidly to concave valley and reduce the height difference between peak and valley, thus reducing the sample surface roughness value [17]. The static pressure determines the surface impact degree of the sample. With the increase in static pressure, the impact degree of the tool head on the sample surface increases, resulting in the peak being rolled flat to a greater extent and filled into the valley. Therefore, the increase in static pressure reduces the roughness value and the roughness reduction rate increases (Figure 4). The increase in the impact degree on the sample surface also leads to the intensification of the surface deformation and the obvious refining of the microstructure. Therefore, the static pressure increases, the surface hardness and residual stress of the sample increases (Figure 6 and Figure 7), and the proportion of high-angle boundaries also increases (Figure 9), among which, nano-equiaxed grains are formed on the surface of SUR2 (Figure 10a).

Because the surface ultrasonic rolling tool head impact frequency is fixed, the number of tool head impacts on the sample surface is constant. The feed rate and the rotational speed determine the continuity of the rolling direction on the sample surface. When the feed rate and rotational speed are large, the degree of discontinuity on the specimen surface increases, and the local position of the sample surface is not processed. Therefore, the increase of the feed rate and rotational speed will increase the roughness value and reduce the roughness reduction rate (Figure 4). At the same time, the surface deformation of the sample is weakened, and the hardness and residual stress increase less (Figure 6, Figure 7, Figure 8, Figure 9 and Figure 10).

There is an inseparable relationship between hardness change and microstructure evolution, and the degree of microstructure refinement directly determines the hardness. After surface ultrasonic rolling treatment, obvious plastic deformation occurs on the sample surface, and even nano-equiaxed grains are formed, and the grains are refined (Figure 8, Figure 9 and Figure 10).

In addition, a large number of dislocation movements, dislocation pile-up, and dislocation interaction occur, resulting in work-hardening of the sample surface. Therefore, the maximum hardness is always on the surface under the action of grain refinement and work-hardening (Figure 7a). With the increase in distance from the surface, the degree of microstructure refinement and work-hardening gradually weakens, resulting in the gradual reduction of hardness to the original state (Figure 7b).

### 4.2. Influence of Surface Ultrasonic Rolling on Fatigue Life

The surface roughness of the original sample was large (Figure 4), and there were obvious convex peaks and concave valleys left by turning, resulting in more defects left by surface turning (Figure 5a). Under the action of stress, stress concentration easily occurred at the defect, leading to the initiation of cracks on the surface [18]. Therefore, under cyclic loading of the sample during the fatigue test, the crack of the original sample was initiated on the surface and then propagated in the opposite direction to the crack source (Figure 13a–c). Under the action of cyclic load, the crack propagated continuously, causing most of the material to fracture. However, in the unfractured part of the material, the crack propagated rapidly under the action of tensile and compressive stresses, resulting in the instantaneous fracture of the material. Therefore, the surface of the material often had obvious dimple morphology (Figure 13d). There were plastic and brittle fractures in the fracture region because of the severe plastic deformation of the fracture instantaneously. In a word, the larger roughness value, lower hardness, and residual compressive stress of the original sample surface made the sample prone to fatigue under lower loads or have a lower life under larger loads (Figure 11 and Figure 13).

After surface ultrasonic rolling under different processes, the thickness of the sample surface decreased significantly and the surface became flat. The surface had obvious plastic deformation, the microhardness increased significantly, and the surface defects were reduced to a great extent. The surface residual stress also observably increased (Figure 4, Figure 5, Figure 6, Figure 7, Figure 8, Figure 9 and Figure 10). These combined factors led to crack initiation on the secondary surface [19]. The increase in residual stress improved the deformation resistance energy of the material and thus the matrix strength [20]. According to literature [21], residual stress can effectively inhibit crack initiation and propagation. Moreover, the crack growth rate can be reduced [22]. The existence of a plastic deformation layer makes the surface material hardened and the surface grain refined, which increases the yield strength of the material and hinders the crack propagation. Therefore, after surface ultrasonic rolling treatment, the fatigue resistance and fatigue life of the sample were improved. The fatigue fracture of the sample required greater stress and longer cycle numbers (Figure 11 and Figure 12), and the fatigue life was increased. In the S-N curve, the fatigue strength of the sample with surface ultrasonic rolling treatment was significantly higher than that of the original sample.

## 5. Conclusions

The surface of TC4 titanium alloy was strengthened and modified by changing the surface ultrasonic rolling process, and its influence on the fatigue properties of titanium alloy was studied. The surface roughness, microhardness, residual stress, and microstructure of TC4 titanium alloy were measured before and after surface ultrasonic rolling; the fatigue properties of TC4 titanium alloy samples were tested after different surface ultrasonic rolling processes; the fatigue fracture characteristics of the samples were analyzed; and the surface strengthening and modification mechanisms of the titanium alloy were discussed. The following conclusions are drawn:Under different process parameters, the surface roughness of titanium alloy samples decreased, and the roughness reduction rate increased. With the increase in static pressure, the surface roughness decreased. With the increase in feed rate and rotational speed, the roughness difference increased and the roughness reduction rate decreased.The surface hardness and residual compressive stress of the samples increased in different degrees. With the increase in static pressure, the surface hardness and compressive residual stress increased. With the increase in feed rate and rotational speed, the surface hardness and residual stress decreased, and the hardness and residual stress of SUR 2 were the highest, increasing by 35%. The axial compressive residual stress increased from −260 MPa to −685 MPa.After surface ultrasonic rolling treatment, obvious plastic deformation occurred on the surface of the titanium alloy sample. The thickness of the obvious plastic deformation layer was about 40 μm at most, and twins appeared in the surface grains of the sample. After surface ultrasonic rolling, there was obvious grain refinement in the surface layer and the number of large grain boundaries. There were approximately 1.5 μm-thick nano-equiaxed grains in the surface layer.Surface roughness, residual compressive stress, and surface hardness were all factors that improved fatigue life. The fatigue strength increased by 52%, from 280 MPa to 425 MPa. Under 450 MPa, the fatigue life of SUR 2 was the highest, at about 7.7 times that of the original. In addition, the crack source was located in the subsurface under the strengthened layer when the fatigue fracture occurred in the sample after surface ultrasonic rolling. The crack source was located on the surface of the sample surface when the fatigue fracture occurred in the sample without surface ultrasonic rolling. The surface integrity of titanium alloy samples after surface ultrasonic rolling treatment was greatly improved, which was the reason for the significant increase of fatigue life of the samples.

## Figures and Tables

**Figure 1 materials-16-00485-f001:**
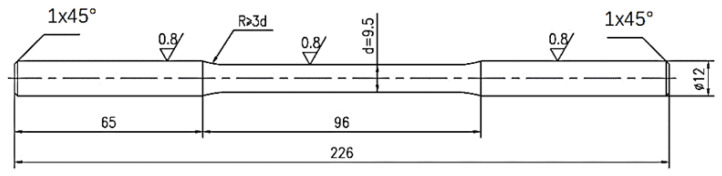
Size diagram of fatigue sample.

**Figure 2 materials-16-00485-f002:**
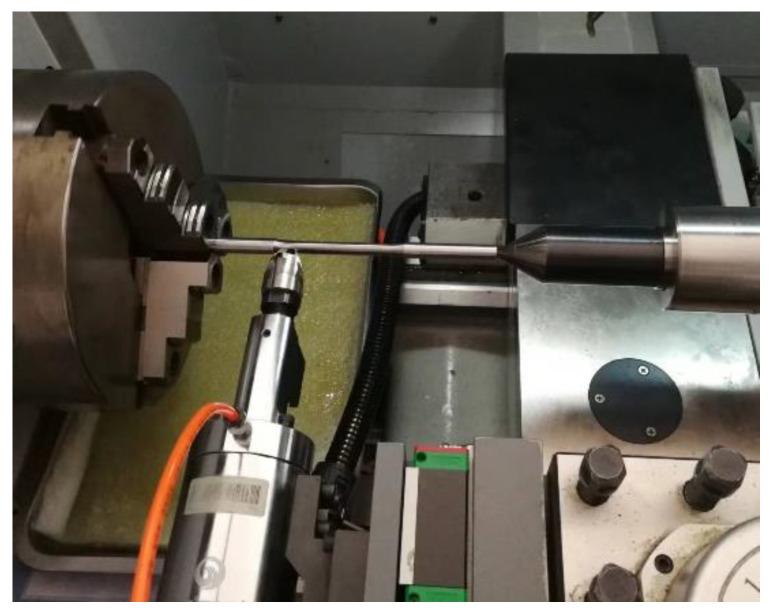
Surface ultrasonic rolling treatment of sample.

**Figure 3 materials-16-00485-f003:**
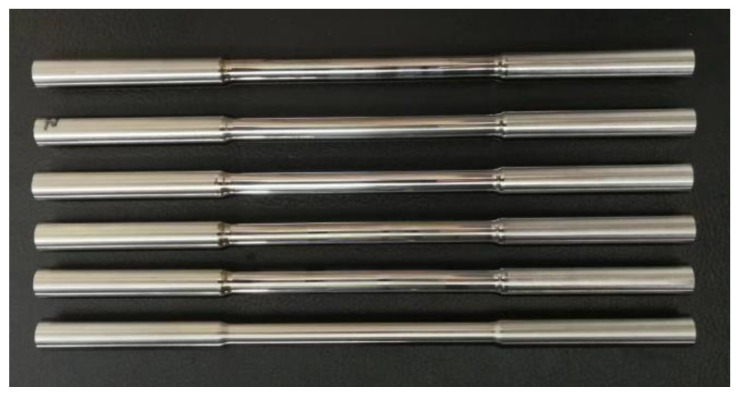
Morphology of samples after surface ultrasonic rolling.

**Figure 4 materials-16-00485-f004:**
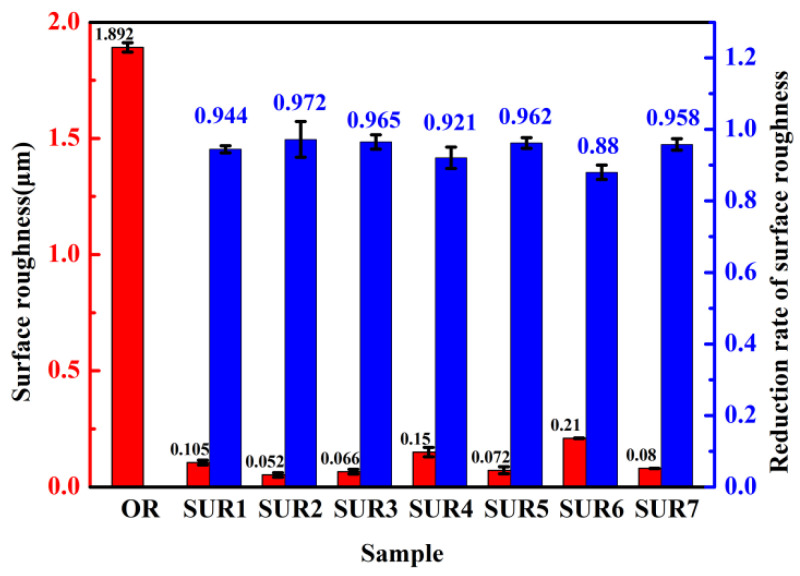
Surface roughness and roughness reduction rate before and after rolling.

**Figure 5 materials-16-00485-f005:**
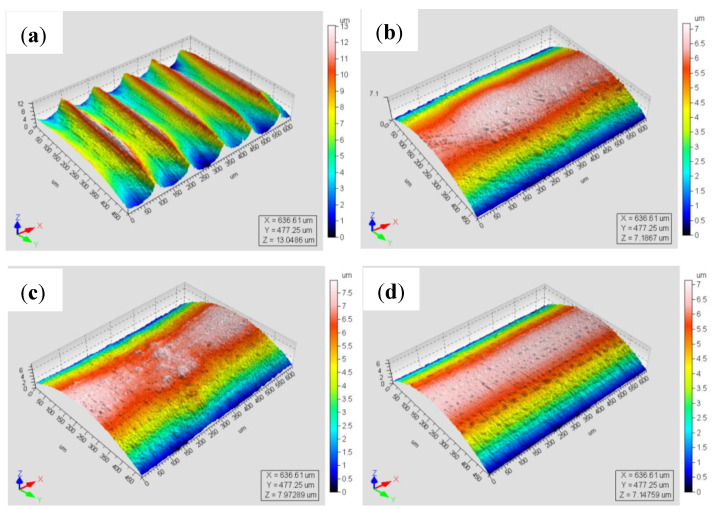
Surface 3D morphologies of different surface ultrasonic rolling processes. (**a**) Original surface 3D morphologies; (**b**–**h**) the surface 3D morphologies of process SUR-SUR7 samples, respectively.

**Figure 6 materials-16-00485-f006:**
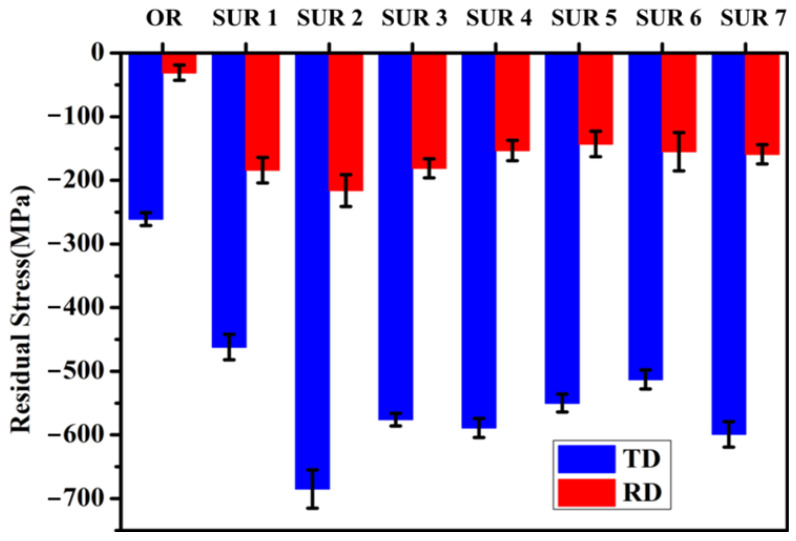
Surface residual stress changes of samples under different surface ultrasonic rolling treatments.

**Figure 7 materials-16-00485-f007:**
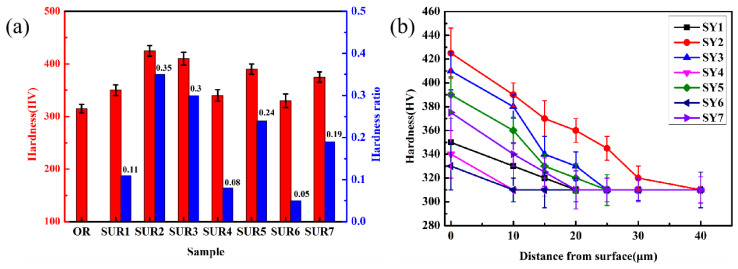
Hardness changes under different surface ultrasonic rolling processes. (**a**) Surface hardness change, (**b**) cross-sectional hardness change.

**Figure 8 materials-16-00485-f008:**
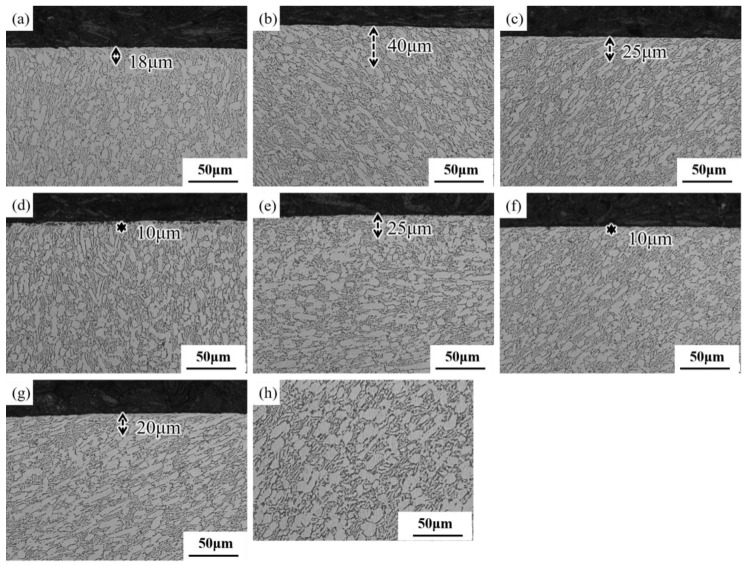
Microstructural changes under different surface ultrasonic rolling processes. (**a**–**g**) SUR 1–SUR 7, (**h**) original microstructure.

**Figure 9 materials-16-00485-f009:**
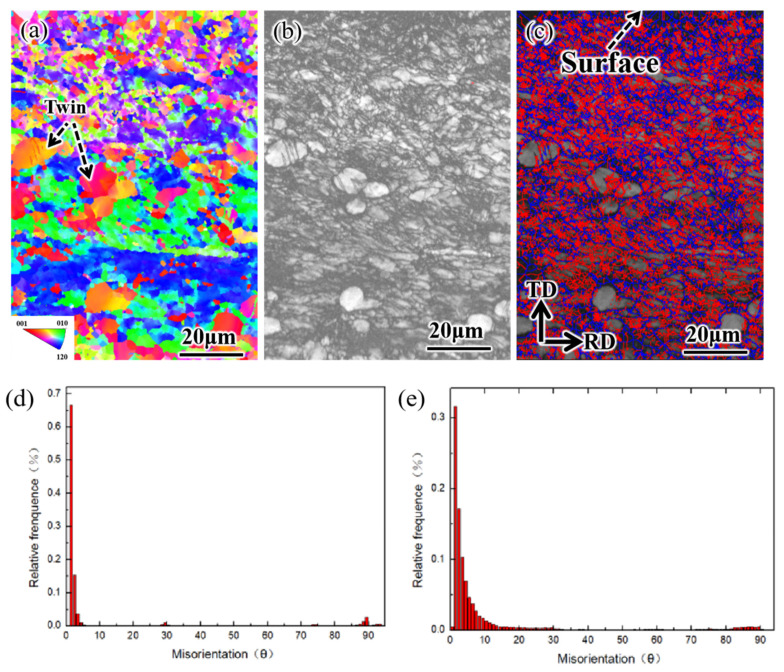
EBSD characterization of microstructure after surface ultrasonic rolling. (**a**) IPF map, (**b**) BC map, (**c**) boundary orientation map (red line > 10°, blue line 2–10°), (**d**) matrix boundary angle distribution, (**e**) surface boundary angle distribution.

**Figure 10 materials-16-00485-f010:**
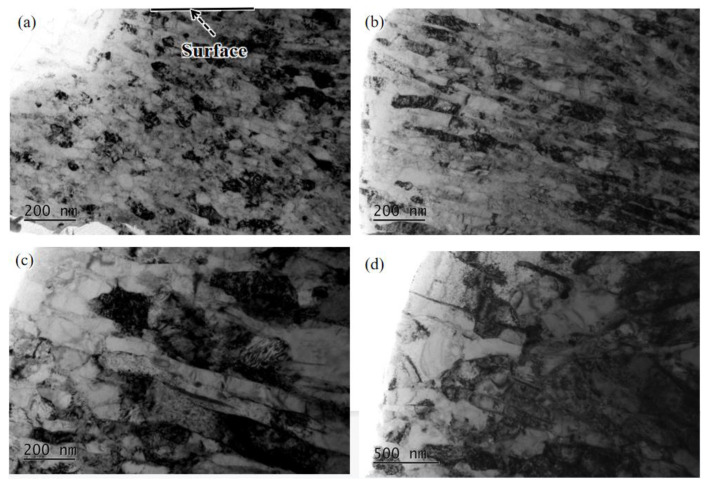
Bright field images of microstructural changes of SUR 2 samples at different distances. (**a**) 0–1.5 μm, (**b**) 10 μm, (**c**) 15 μm, (**d**) 20 μm.

**Figure 11 materials-16-00485-f011:**
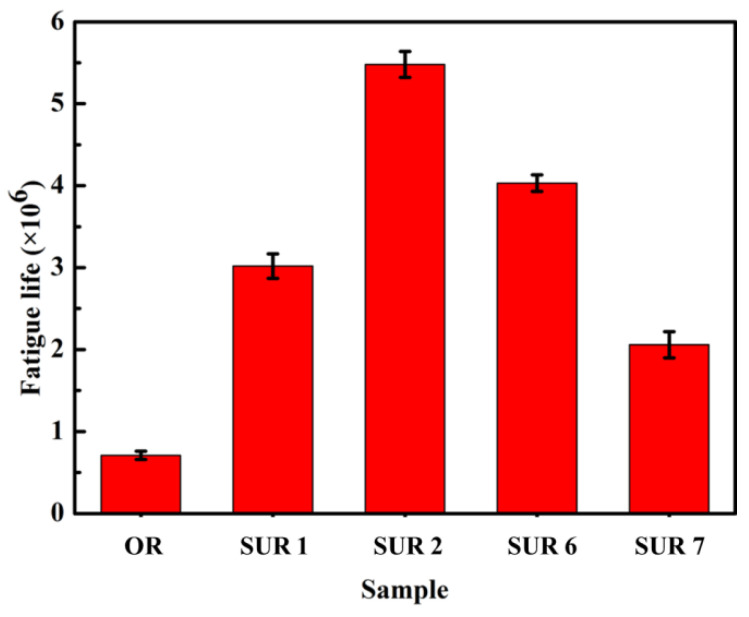
Fatigue life changes.

**Figure 12 materials-16-00485-f012:**
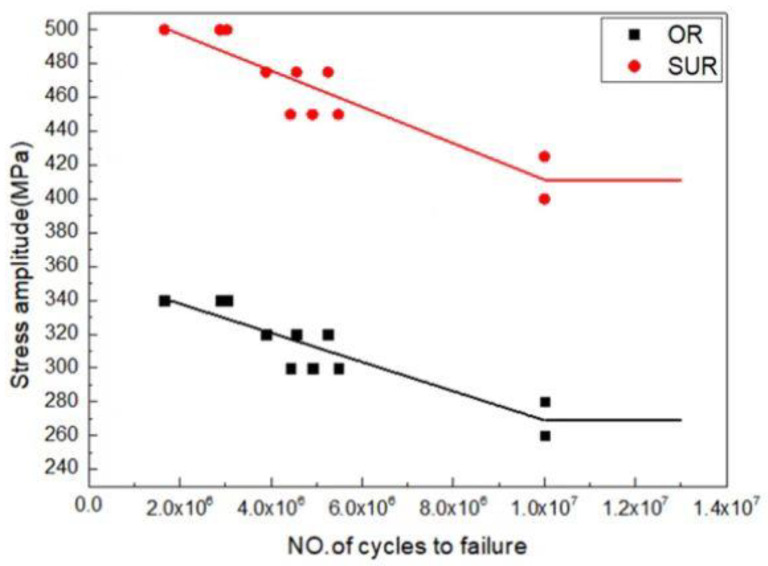
S-N curves of the original and SUR 2 samples.

**Figure 13 materials-16-00485-f013:**
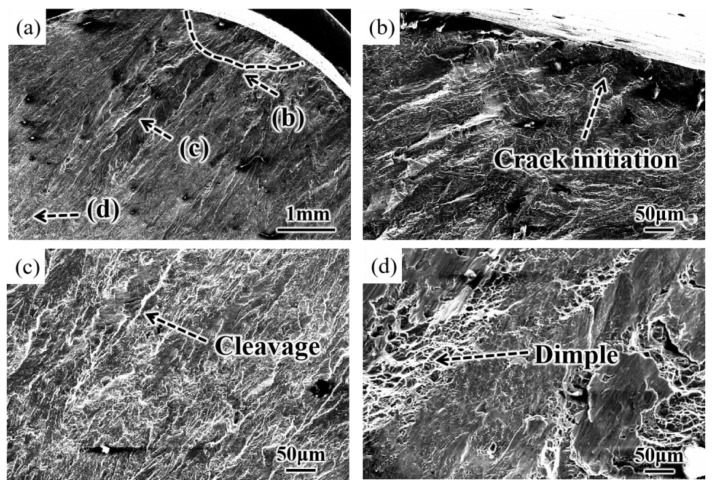
Fracture morphology of original sample. (**a**) Low multiplier, (**b**) crack source region, (**c**) crack propagation region, (**d**) fracture region.

**Figure 14 materials-16-00485-f014:**
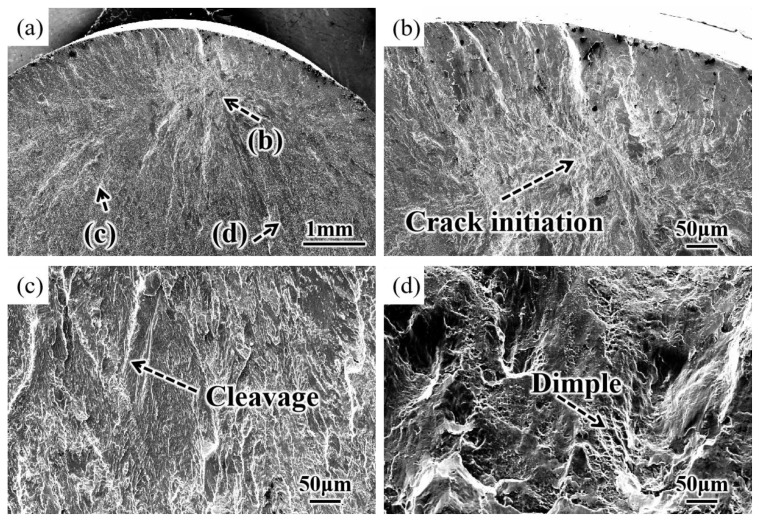
Fracture morphology of SUR2 sample. (**a**) Low multiplier, (**b**) crack source region, (**c**) crack propagation region, (**d**) fracture region.

**Table 1 materials-16-00485-t001:** Main chemical components of TC4 titanium alloy.

Al	V	Fe	C	O	N	H	Ti
6.0	4.0	0.12	0.02	0.09	0.01	0.002	89.758

**Table 2 materials-16-00485-t002:** Parameters of surface ultrasonic rolling treatment using single-factor variable method.

Test Number	Rational Speed (r/min)	Feed Rate (mm/r)	Static Pressure (kg)
SUR 1	70	0.10	30
SUR 2	70	0.10	90
SUR 3	70	0.05	60
SUR 4	70	0.15	60
SUR 5	28	0.10	60
SUR 6	132	0.10	60
SUR 7	70	0.10	60

## Data Availability

Not applicable.

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
