# Peer review of "Study on Surface Integrity and Fatigue Properties of TC4 Titanium Alloy by Surface Ultrasonic Rolling"

_materials, 2023, doi:10.3390/ma16020485_

Round 1

Reviewer 1 Report

The present work is devoted to the study of the surface integrity of titanium alloys after ultrasonic rolling. The work is extremely interesting and the authors perform large amount of experiments. Considering this I strongly suggest publication in Materials after the following  revisions.

-Please improve the introduction section adding and discussing more references regarding surface integrity of titanium such as  https://doi.org/10.1007/s11665-018-3598-x and ultrasonic-assisted machining such as https://doi.org/10.1016/j.cirp.2021.04.040. The low number of references is in fact in my opinion the main lack of the present work

-Please state clearly in the introduction section the novelty of the present work in comparison with the literature

-Please add in the experimental section more details regarding the preparation of the samples for SEM and TEM observation (metallographic procedure) and about the parameters employed in the XRD residual stresses evaluation

-Please add error bars in Fig.4

Author Response

Response to Reviewer 1

Dear reviewer,

First, I am very grateful for your comments and suggestions regarding the manuscript entitled “Study on surface integrity and fatigue properties of TC4 titanium alloy by surface ultrasonic rolling”. The comments were all valuable and very helpful in revising and improving our paper, and they also provided important guidance for our research. We studied the comments carefully and made corrections, which we hope will meet with approval. The main corrections to the paper and the responses to the editors’ and reviewers’ comments are as follows:

Responds to the reviewer’s comments:

  1. Please improve the introduction section adding and discussing more references regarding surface integrity of titanium such as  https://doi.org/10.1007/s11665-018-3598-x and ultrasonic-assisted machining such as https://doi.org/10.1016/j.cirp.2021.04.040. The low number of references is in fact in my opinion the main lack of the present work
  2. Please state clearly in the introduction section the novelty of the present work in comparison with the literature

Response: 

1 Introduction

Titanium alloy is an important structural material, with high strength, low density and excellent corrosion resistance, and easy to meet the requirements of lightweight, which is widely used in automotive, aerospace, and biomedical and other fields[1-3]. However, they have the disadvantages of low hardness, low wear resistance and low fatigue resistance [4-6]. If it is light, parts will be damaged, and if it is heavy, major safety accidents will occur, which limits the application range of titanium alloy.

The surface machining technology of titanium alloy has been studied extensively. Sartori studied the integrity of the semi-finished surface of Ti6Al4V titanium alloy under different cryogenic cooling techniques, and the results showed that the machined surface had no defects and had the best integrity when N2 cooling was used at -150 ℃ [7]. Lin treated the surface of titanium alloy with sandblasting and grinding methods and carried out 3-point bending test. The results showed that the Ti-ceramic bond strength of sandblasting samples was higher than that of grinding samples [8]. Gupta used plasma spraying technology to prepare TiN coating on Ti6Al4V surface in situ, and the results showed that the coating showed unusually high mechanical properties. After spraying, the wear rate of Ti6Al4V was significantly reduced by about 15 times, and the corrosion rate was significantly reduced by about 4 times [9]. lv conducted ultrasonic deep rolling (UDR) treatment on the surface of titanium alloy, and studied the influence of UDR on the surface morphology and surface roughness, showing that UDR treatment can significantly reduce machining marks and greatly reduce the surface roughness [10]. Li used shot peening to carry out surface strengthening treatment on TC11 titanium alloy, showing that shot peening can introduce residual compressive stress field with thickness of about 230 MPa on the surface layer of the sample [11]. Xue adopted the method of micro-laser shock strengthening to strengthen the TC17 titanium alloy, and the results showed that the fatigue strength of the sample strengthened increased by 32% [12]. Kattoura used ultrasonic nanocrystal surface modification (UNSM) technology to surface process ATI 718 Plus. The results showed that after UNSM, nano-sized grains, twins and high fault density were generated in the near-surface region of the sample, and the increase of surface hardness and residual stress significantly reduced the crack growth rate [13]. Ren used different ultrasonic surface rolling (USR) processes to roll Ti5Al4Mo6V2Nb1Fe titanium alloy. The results showed that serious plastic deformation occurred on the sample surface, and the surface roughness, hardness and residual stress were optimized. After USR, the wear rate was reduced by about 70%, and the fretting wear resistance was improved [14]. Kumar used ultrasonic shot peening to treat the surface of titanium alloy, and the results showed that a deformation layer of about 30 μm was formed on the surface of the sample, with a grain size of 17-25 nm and significant refinement [15].

To sum up, most scholars use low-temperature cooling, shot peening, ultrasonic surface rolling and other technologies to treat the surface of titanium alloy, so as to improve surface properties, while the use of surface ultrasonic rolling technology to study the surface integrity and microstructure of titanium alloy changes is not deep enough. In this paper, surface ultrasonic rolling process was used to study TC4 titanium alloy surface ultrasonic rolling. The influence of ultrasonic rolling process parameters on surface properties was studied by changing static pressure, feed rate and rational speed, and the influence rule on fatigue properties of titanium alloy materials was studied. The surface modification and fatigue property of TC4 titanium alloy were studied in depth, and the reference basis and theoretical support for the anti-fatigue manufacturing of TC4 titanium alloy were hoped to be provided.

"Https://doi.org/10.1007/s11665-018-3598-x, topic is: Surface integrity analysis of Ti6Al4V after semi-finishing turning under different low-temperature cooling strategies" this literature the author cited in the references [7]. "Https://doi.org/10.1016/j.cirp.2021.04.040, topic is: Surface texturing to enhance sol-gel coating performances for biomedical applications" this literature is excellent, the author carefully read, benefit, but this article is to discuss the surface of magnesium alloy processing, and the content difference is bigger, so the authors have no reference, I'm very sorry.

[7] S. Sartori, L. Pezzato, M. Dabala`, T. Maurizi Enrici, A. Mertens, A. Ghiotti, S. Bruschi, Surface integrity analysis of Ti6Al4V after semi-finishing turning under different low-temperature cooling strategies. Journal of Materials Engineering and Performance, 2018, 27.

  1. Please add in the experimental section more details regarding the preparation of the samples for SEM and TEM observation (metallographic procedure) and about the parameters employed in the XRD residual stresses evaluation.

Response: The residual stress on the surface of the samples is measured by i-XRD X-ray stress tester. The type of tube ball is Co, the test voltage is 25 kV, the operating current is 4.5 mA, and the diffraction angle is 155 °. The angle instrument is used for accurate measurement, the exposure time is 1 s, the spot size is 2 mm×1 mm, the diffraction crystal plane is (114) crystal plane, the scanning range of bragg angle 2θ is 120Ëš ~170Ëš, and the peak location method is Gaussian.

2.2 SEM, OM, EBSD and TEM sample preparation

The fractured sample is cut into small samples with a height of 20 mm and a diameter of 9.5 mm along the fracture direction. Petroleum ether and alcohol are respectively used for ultrasonic cleaning in ultrasonic equipment (SB-5200DTN) to prevent oxidation of the fracture. Qualified fracture samples are prepared, and then the fracture morphology is observed by SEM.

All OM and EBSD samples are cut into 8 mm × 8 mm × 3 mm along the rolling direction and then progressively polished with 400-1200 grit SiC sandpaper [16]. Finally, the samples are polished with 10 μm, 5 μm, 3 μm and 1μm diamond paste respectively for 6 min, 5 min, 5 min and 5 min. The corrosion solution is prepared by 5% HF+10% HNO3+85% H2O (distilled water), and the polished samples are etched to prepare qualified OM samples. The GATAN Ilion II 697 argon ion polishing equipment is used to polish the sample surface with argon ion, the voltage is 5 kV and the polishing time is 4 h, and qualified EBSD samples are prepared.

TEM sample preparation is to use a cutting machine to cut a 3.2 mm diameter semi-cylinder from the section of the sample, use a low-speed saw to cut it into a 1 mm thickness semi-circular slice, polish it into a 35-40 μm semi-circular slice with different types of sandpaper, and then use Gatan PIPS 691 ion thinning instrument for ion thinning of the slice. TEM samples are prepared.

  1. Please add error bars in Fig.4

Response: 

Figure 4. Surface roughness and roughness reduction rate before and after rolling

We tried our best to improve the manuscript and made some changes in  the manuscript. These changes will not  influence  the  content  and  framework  of the paper. And here we did not list the changes but marked  in  cyan  in revised paper. 

We appreciate for Editors/Reviewers’ warm  work  earnestly, and hope that the correction will meet with approval.

Thanks again for your suggestions. I hope I can learn more from you.

Reviewer 2 Report

1. Please consider reviewing the abstract and highlight the novelty, major findings and conclusions.

2. Authors should explain the main theoretical and practical contributions of this study to this research field.

3. The results are merely discussed and is limited to comparing the experimental observation. The authors are encouraged to include a discussion section and critically discuss the observations from this investigation with existing literature.

4. Table 1 is repeated twice. Check the sequence.

Author Response

Response to Reviewer 2

Dear reviewer,

First, I am very grateful for your comments and suggestions regarding the manuscript entitled “Study on surface integrity and fatigue properties of TC4 titanium alloy by surface ultrasonic rolling”. The comments were all valuable and very helpful in revising and improving our paper, and they also provided important guidance for our research. We studied the comments carefully and made corrections, which we hope will meet with approval. The main corrections to the paper and the responses to the editors’ and reviewers’ comments are as follows:

Responds to the reviewer’s comments:

  1. Please consider reviewing the abstract and highlight the novelty, major findings and conclusions.

Response: Abstract: In this paper, the influence of surface ultrasonic rolling process on the surface integrity of TC4 titanium alloy and its influence on the fatigue properties were studied. By comparing and analyzing the surface roughness, microhardness, residual stress, microstructure and fatigue fracture, the surface strengthening and modification mechanism of TC4 titanium alloy was discussed. The results show that the surface roughness of titanium alloy is decreased obviously after the suitable surface ultrasonic rolling process, and the maximum Ra value can be reduced to 0.052 μm. The axial residual stress on the specimen surface can be increased to -685 MPa. The hardening rate of the surface hardness of the sample is 35%. The residual compressive stress and hardness of the sample surface increase with the increase of static pressure. However, the increase of feed rate and rational speed is less. After surface ultrasonic rolling, the sample surface has obvious grain refinement, the number of high-angle boundaries increases, and even the formation of nano-equiaxed grains. The fatigue strength increased by 52% from 280 MPa to 425 MPa. Under 450 MPa, the fatigue life of samples with SUR 2 is the highest, which is about 7.7 times of the original samples. The surface integrity of titanium alloy samples after surface ultrasonic rolling treatment is greatly improved, which is the reason for the significant increase of fatigue life of the samples.

  1. Authors should explain the main theoretical and practical contributions of this study to this research field.

Response: Most scholars use low-temperature cooling, shot peening, ultrasonic surface rolling and other technologies to treat the surface of titanium alloy, so as to improve surface properties, while the use of surface ultrasonic rolling technology to study the surface integrity and microstructure of titanium alloy changes is not deep enough. In this paper, surface ultrasonic rolling process was used to study TC4 titanium alloy surface ultrasonic rolling. The influence of ultrasonic rolling process parameters on surface properties was studied by changing static pressure, feed rate and rational speed, and the influence rule on fatigue properties of titanium alloy materials was studied. The surface modification and fatigue property of TC4 titanium alloy were studied in depth, and the reference basis and theoretical support for the anti-fatigue manufacturing of TC4 titanium alloy were hoped to be provided.

  1. The authors are encouraged to include a discussion section and critically discuss the observations from this investigation with existing literature.

Response: 

4 Discussion

4.1 Influence of surface ultrasonic rolling on surface integrity

In the process of surface ultrasonic rolling treatment, the static pressure, feed rate and rotational speed directly determine the surface machining degree of the sample. After surface ultrasonic rolling treatment, the metal on the surface of the sample will undergo plastic deformation, which will prompt the material with convex peak to flow rapidly to concave valley, and reduce the height difference between peak and valley, thus reducing the sample surface roughness value [17]. The static pressure determines the surface impact degree of the sample. With the increase of static pressure, the impact degree of the tool head on the sample surface increases, resulting in the peak being rolled flat to a greater extent and filled into the valley. Therefore, the increase of static pressure reduces the roughness value and the roughness reduction rate increases (Figure 4). The increase of the impact degree on the sample surface also leads to the intensification of the surface deformation and the obvious refining of the microstructure. Therefore, the static pressure increases, the surface hardness and residual stress of the sample increases (Figure 6, Figure 7), and the proportion of high-angle boundaries also increases (Figure 9), among which nano-equiaxed grains are formed on the surface of SUR2 (Figure 10a).

Because the surface ultrasonic rolling tool head impact frequency is fixed, so the number of tool head impact on the sample surface is constant. The feed rate and the rotational speed determine the continuity of the rolling direction on the sample surface. When the feed rate and rotational speed are large, the degree of discontinuity on the specimen surface increases, the local position of the sample surface is not processed. Therefore, the increase of the feed rate and rotational speed will increase the roughness value and reduce the roughness reduction rate (Figure 4). At the same time, the surface deformation of the sample is weakened, and the hardness and residual stress increase less (Figures 6-10).

There is an inseparable relationship between hardness change and microstructure evolution, and the degree of microstructure refinement directly determines the hardness. After surface ultrasonic rolling treatment, obvious plastic deformation occurs on the sample surface, and even nano-equiaxed grains are formed, and the grains are refined (Figures 8-10).

In addition, a large number of dislocation movements, dislocation pile-up and dislocation interaction, resulting in work hardening of the sample surface. Therefore, the maximum hardness is always on the surface under the action of grain refinement and work hardening (Figure 7a). With the increase of distance from the surface, the degree of microstructure refinement and work hardening gradually weakens, resulting in the gradual reduction of hardness to the original state (Figure 7b).

4.2 Influence of surface ultrasonic rolling on fatigue life

The surface roughness of the original sample is large (Figure 4), and there are obvious convex peaks and concave valleys left by turning, resulting in more defects left by surface turning (Figure 5a). Under the action of stress, stress concentration is easy to occur at the defect, leading to the initiation of cracks on the surface [18]. Therefore, under cyclic loading of the sample during the fatigue test, the crack of the original sample is initiated on the surface and then propagated in the opposite direction to the crack source (Figures 13a-c). Under the action of cyclic load, the crack propagates continuously, causing most of the material to fracture. However, in the unfractured part of the material, the crack propagates rapidly under the action of tensile and compressive stresses, resulting in the instantaneous fracture of the material. Therefore, the surface of the material often has obvious dimple morphology (Figure 13d). There are plastic and brittle fractures in the fracture region because of the severe plastic deformation of the fracture instantaneously. In a word, the larger roughness value, lower hardness and residual compressive stress of the original sample surface make the sample prone to fatigue under lower loads or have a lower life under larger loads (Figure 11, Figure 13).

After surface ultrasonic rolling under different processes, the thickness of the sample surface decreases significantly and the surface becomes flat. The surface has obvious plastic deformation, the microhardness increases significantly, and the surface defects are reduced to a great extent. The surface residual stress also increases obviously (Figures 4-10). These combined factors lead to crack initiation on the secondary surface [19]. The increase of residual stress improves the deformation resistance energy of the material and thus the matrix strength [20]. According to literature [21], residual stress can effectively inhibit crack initiation and propagation. Moreover, the crack growth rate can be reduced [22]. The existence of plastic deformation layer makes the surface material hardened and the surface grain refined, which increases the yield strength of the material and hinders the crack propagation. Therefore, after surface ultrasonic rolling treatment, the fatigue resistance and fatigue life of the sample are improved. The fatigue fracture of the sample requires greater stress and longer cycle numbers (Figure 11, Figure 12), and the fatigue life is increased. In the S-N curve, the fatigue strength of the sample with surface ultrasonic rolling treatment is significantly higher than that of the original sample.

  1. Table 1 is repeated twice. Check the sequence.

Response: The form has been corrected.

We tried our best to improve the manuscript and made some changes in  the manuscript. These changes will not  influence  the  content  and  framework  of the paper. And here we did not list the changes but marked  in  cyan  in revised paper. 

We appreciate for Editors/Reviewers’ warm  work  earnestly, and hope that the correction will meet with approval.

Thanks again for your suggestions. I hope I can learn more from you.

Round 2

Reviewer 1 Report

The paper Is now acceptable for publication